# A Bayesian decision support system for automated insulin doses in adults with type 1 diabetes on multiple daily injections: a randomized controlled trial

Alessandra Kobayati [1,2], Anas El Fathi [3,4], Natasha Garfield[2,5], Laurent Legault[2,6], Adnan Jafar [2,3], Jean-François Yale [2,5], Michael A. Tsoukas[2,5,7] & Ahmad Haidar [1,2,3,5,7] ✉

Achieving optimal glycemic control remains challenging for many individuals with type 1 diabetes using multiple daily injections. We report results from a 12-week, open-label, randomized controlled trial evaluating a decision support system (DSS) consisting of a mobile application and a titration algorithm that provides weekly basal and prandial insulin recommendations. Eighty-four adults with type 1 diabetes and suboptimal glycemic control (HbA1c ≥ 7.5%) are randomized 1:1 to receive the DSS or a non-adaptive bolus calculator (control), alongside Freestyle Libre glucose sensors. The primary endpoint is change in HbA1c from baseline; secondary endpoints include additional glycemic and insulin-related metrics. The DSS reduces mean HbA1c from 8.6% (SD 1.1) to 8.1% (0.8) (p = 0.0002), while the control reduces HbA1c from 8.6% (1.0) to 8.5% (1.0) (p = 0.22); yielding a treatment effect of −0.40% (95% CI: −0.75 to −0.051; p = 0.025). There are no reported severe hypoglycemia or diabetic ketoacidosis events. Our DSS improves HbA1c in this population without compromising safety. ClinicalTrials.gov: NCT04123054.

Type 1 diabetes is characterized by the destruction of pancreatic β cells, requiring lifelong insulin replacement therapy[1]. Intensive insulin therapy, through multiple daily injections (MDI) or an insulin pump, with a target glycated hemoglobin (HbA1c) < 7%, reduces microvascular and macrovascular complications[2,3]. MDI therapy is used by the majority of individuals with type 1 diabetes worldwide, but it is associated with frequent glycemic fluctuations, primarily due to variability in subcutaneous insulin absorption[4] and insulin sensitivity, physical activities[5], different food composition[6], and stress, among other factors. Furthermore, infrequent clinical follow-ups impede timely adjustments of insulin doses, further exacerbating the challenges with MDI therapy[7]. Most people with type 1 diabetes on MDI therapy do not achieve glycemic targets[8].

The introduction of continuous glucose monitoring (CGM) systems has revolutionized type 1 diabetes care. In the case of insulin pumps, algorithm-guided titrations of insulin doses based on real-time CGM readings are now commercially available and improve glycemic outcomes and quality of life compared to conventional pump

[1]Division of Experimental Medicine, Department of Medicine, McGill University, Montreal, QC, Canada. [2]Research Institute of the McGill University Health Center, McGill University, Montreal, QC, Canada. [3]Department of Biomedical Engineering, McGill University, Montreal, QC, Canada. [4]Center for Diabetes Technology, University of Virginia, Charlottesville, VA, US. [5]Division of Endocrinology, Department of Medicine, McGill University Health Center, Montreal, QC, Canada. [6]Department of Pediatrics, Montreal Children's Hospital, Montreal, QC, Canada. [7]These authors jointly supervised this work: Michael A. Tsoukas, Ahmad Haidar. ✉e-mail: ahmad.haidar@mcgill.ca

**Fig. 1 | Flow diagram of participant recruitment.** CONSORT flow chart showing the number of participants screened, enrolled, allocated to each arm, and included in the analysis. Between March 5, 2020, and September 27, 2023, 179 individuals were pre-screened for interest and basic eligibility; 94 underwent formal screening, of whom 9 were ineligible and 1 withdrew prior to randomization. A total of 84 participants (2 referred from the Jewish General Hospital site) were randomized in a 1:1 ratio to the experimental arm (McGill DSS) or control arm (non-adaptive iBolus app). The primary analysis was conducted on participants who completed at least 10 weeks of intervention with an end-of-study HbA1c measurement.

therapy[9]. For MDI therapy, the recent availability of smart insulin pens and caps[10], which track insulin doses and communicate wirelessly with smartphones, have paved the way for the development of decision support systems (DSSs) that enable algorithm-guided automatic titrations of insulin doses based on retrospective analysis of CGM readings[11]. No such DSS is commercially available.

Several feasibility studies have assessed DSSs for MDI therapy[12–16], but only three randomized controlled trials have been reported so far. The InControl Advice system was tested in a 12-week parallel trial in 80 adolescents and adults[17], the ABC4D system was tested in a 12-week crossover trial in 37 adults[18], and the PEPPER system was tested in a 12-week crossover trial in 54 adults[19]. None of the studies demonstrated improvements in glycemic outcomes with their DSS.

Here, we present the effectiveness of the McGill DSS, which uses a Bayesian optimization algorithm, in 84 adults on MDI therapy with suboptimal glycemic control in a 12-week, open-label, randomized controlled parallel trial.

## Results

This randomized controlled trial was completed prior to analysis. Between March 05, 2020, and September 27, 2023, 179 individuals were pre-screened for interest and basic eligibility. A total of 94 adults on MDI therapy were officially screened; nine were ineligible and one withdrew prior to randomization due to commitment concerns. Consequently, 84 participants with a suboptimal baseline HbA1c level of 7.5% or higher were enrolled in the study. Of these, two participants

were referred from the Jewish General Hospital site and enrolled at the main study site, the McGill University Health Center–Royal Victoria Hospital. Participants were randomized in a 1:1 ratio and stratified according to previous sensor use to receive either the McGill DSS (iBolus app with the optimization algorithm; experimental arm) or the non-adaptive iBolus app (control arm). The recruitment flow of participants is presented in Fig. 1. Baseline characteristics were comparable across both groups (Table 1), including the programmed glucose target. The overall mean age was 38 (12) years, diabetes duration was 22 (12) years, HbA1c was 8.6% (1.1), 44% were female, and 76% were regular sensor users (defined as at least three consecutive months of sensor wear prior to enrollment).

### Primary outcome

The primary endpoint, which was the mean difference in HbA1c from baseline to the end-of-study visit, was analyzed using data from all 84 randomized participants for whom at least 10 completed weeks of intervention were completed. The McGill DSS reduced mean HbA1c from 8.6% (SD 1.1) at baseline to 8.1% (0.8) at end-of-study, while the standalone app reduced mean HbA1c from 8.6% (1.0) to 8.5% (1.0) (Fig. 2). The mean (SD) within-group change in HbA1c from baseline to end-of-study was greater for the experimental group (−0.55% (0.87); p = 0.0002) compared to the control group (−0.14% (0.74); p = 0.22) (Table 2; Supplementary Fig. 1), with a between-group difference of −0.40% (95% CI −0.75 to −0.051; p = 0.025; Table 2; Fig. 2).

## Table 1 | Baseline characteristics of participants (n = 84)

| | Experimental arm (n = 42) | Control arm (n = 42) |
|---|---|---|
| Age (years) | 39 (12) | 38 (12) |
| Female sex, n (%) | 20 (48) | 17 (40) |
| BMI (kg/m$^2$) | 26 [23–29] | 26 [24–29] |
| Duration of diabetes (years) | 21 (11) | 23 (13) |
| HbA1c (%) | 8.6 (1.1) | 8.6 (1.0) |
| Prior rt/isCGM use*, n (%) | 32 (76) | 32 (76) |
| Ethnic origin, n (%) | | |
| White | 29 (69) | 31 (74) |
| Black | 3 (7.1) | 4 (10) |
| Asian Canadian | 3 (7.1) | 2 (4.8) |
| Latinx | 4 (10) | 1 (2.4) |
| Middle Eastern | 2 (4.8) | 2 (4.8) |
| multi-ethnic | 1 (2.4) | 2 (4.8) |
| Mealtime insulin strategy, n (%) | | |
| Carbohydrate counting | 27 (64) | 24 (57) |
| Fixed dose | 15 (36) | 18 (43) |
| Basal pen increment, n (%) | | |
| 2.0U | 2 (4.8) | 2 (4.8) |
| 1.0U | 38 (90) | 37 (88) |
| 0.5U | 2 (4.8) | 3 (7.1) |
| Bolus pen increment, n (%) | | |
| 1.0U | 32 (76) | 34 (81) |
| 0.5U | 10 (24) | 8 (19) |
| Basal insulin types, n (%) | | |
| Degludec | 22 (52) | 25 (60) |
| Glargine U-300 | 3 (7.1) | 5 (11) |
| Glargine U-100 | 12 (29) | 6 (14) |
| Basaglar | 2 (4.8) | 4 (9.5) |
| Determir | 3 (7.1) | 2 (4.8) |
| Bolus insulin types, n (%) | | |
| Aspart | 16 (38) | 18 (43) |
| Trurapi | 1 (2.4) | 3 (7.1) |
| Lispro | 14 (33) | 12 (29) |
| Admelog | 3 (7.1) | 0 (0.0) |
| Fast-acting aspart | 7 (17) | 7 (17) |
| Glulisine | 1 (2.4) | 2 (4.8) |

Data are presented as mean (SD) or median [IQR] unless stated otherwise.

*rt real-time, is: intermittently scanned; new user was defined as <3 months of uninterrupted use prior to study enrollment, and regular user was defined as ≥3 months.

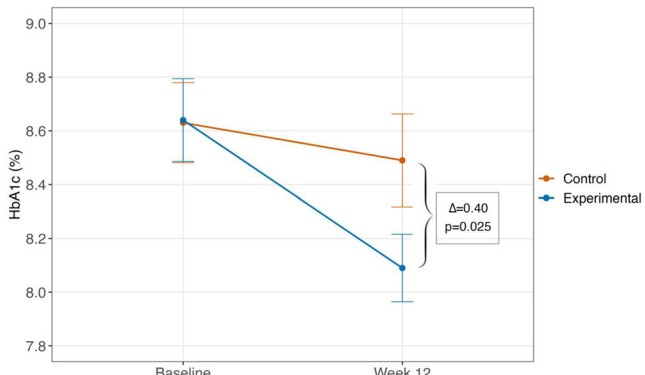

**Fig. 2 | Change in HbA1c from baseline to Week 12.** Mean change in HbA1c (%) from baseline to Week 12 in experimental (n = 42; blue) and control (n = 42; orange) participants. Data are presented as mean (standard error). Between-group differences were assessed using a two-sided linear mixed-effects model adjusting for baseline HbA1c. The treatment effect was −0.40% (95% CI: −0.75 to −0.051; p = 0.025).

## Table 2 | Primary Outcome

| | Experimental (n = 42) | Control (n = 42) | Difference [95% CI] | p value |
|---|---|---|---|---|
| HbA1c change from baseline (%) | −0.55 (0.87) | −0.14 (0.74) | −0.40 [−0.75 to −0.051] | 0.025 |

Data presented as mean (SD) or median [IQR]. A two-sided linear mixed-effects model was used for the analysis.

p = 0.32, respectively; Table 3). Furthermore, HbA1c improvements appeared independent of app usage, as frequency was not associated with better glucose control, as shown in Supplementary Fig. 2. This suggests that app usage was likely driven by meal consumption rather than engagement or adherence. Further post-hoc assessments of app usage revealed that participants in the experimental and control groups used the app to determine prandial bolus doses 2.7 [2.4–3.1] and 2.5 [1.9–3.0] times per day (p = 0.11), respectively, and to determine correction bolus doses (outside of mealtime) 0.56 [0.17–1.3] and 0.46 [0.21–1.2] times per day (p = 0.93), respectively (Supplementary Table 1). Participants also used the app to log their basal dose 0.92 [0.82–0.98] and 0.86 [0.67–0.94] times per day (p = 0.0078), respectively. App usage for bolus dose determination and basal logging remained stable throughout the study (Supplementary Fig. 3).

With respect to the review of algorithm recommendations, there were 50 instances of large accumulative changes (defined as >30% from baseline or a previously approved change) in insulin parameters among participants from the experimental group. When flagged, these changes required review and approval by a member of the study's clinical team prior to being transmitted to participants. All reviewed changes were approved without modification to the algorithm's recommendations. Of the 504 total algorithm-generated recommendations, 41 were overridden by participants in the experimental group, compared to 65 self-initiated or physician-directed parameter changes in the control group, consistent with standard of care.

Other prespecified secondary endpoints included CGM and insulin metrics. There were no differences between the groups in any of the percentages of time spent in, below, or above target glucose ranges, nor in glycemic variability over the 12-week period (Supplementary Table 2). Similar to CGM outcomes, there were no between-group differences in total daily insulin (Supplementary Table 2) or in mean changes (from baseline to the final week) in basal and prandial

### Secondary outcomes and post-hoc analyses

All prespecified secondary outcomes were calculated using data from all 84 participants. The proportion of participants who achieved an end-of-study HbA1c level of ≤7.5% and ≤7.0% was 19% versus 17% (p = 1.0) and 12% versus 0% (p = 0.055) for the experimental and control groups, respectively.

Additional related post-hoc analyses were performed to further explore associations and outcome patterns. In one such analysis of categorical HbA1c changes, a greater proportion of participants in the experimental group achieved HbA1c reductions of ≥0.5%, ≥1.0%, and ≥1.5% compared to the control group (52% vs. 31%, p = 0.048; 19% vs. 10%, p = 0.21; and 12% vs. 5%, p = 0.24, respectively; Table 3). Conversely, the proportions of participants with HbA1c worsening of ≥0.5%, ≥1.0%, and ≥1.5% were lower in the experimental group compared to the control group (2% vs. 21%, p = 0.0074; 2% vs. 7%, p = 0.31; and 0% vs. 2%,

**Table 3 | Proportion of participants with improvements and degradations in HbA1c by ≥0.5%, ≥1.0%, ≥1.5% compared to baseline**

|  | Proportion of experimental participants (%) | Proportion of control participants (%) | Proportion difference (%) [95% CI] | p value |
|---|---|---|---|---|
| HbA1c improvement (%) |  |  |  |  |
| ≥0.5 | 52 | 31 | 21 [0.4 to 40] | 0.048 |
| ≥1.0 | 19 | 10 | 10 [−6 to 25] | 0.21 |
| ≥1.5 | 12 | 5 | 7 [−6 to 21] | 0.24 |
| HbA1c degradation (%) |  |  |  |  |
| ≥0.5 | 2 | 21 | −19 [−33 to −5] | 0.0074 |
| ≥1.0 | 2 | 7 | −5 [−17 to 6] | 0.31 |
| ≥1.5 | 0 | 2 | −2 [−12 to 6] | 0.32 |

A two-sided chi-squared test was used to assess differences in proportions. No adjustments were made for multiple comparisons.

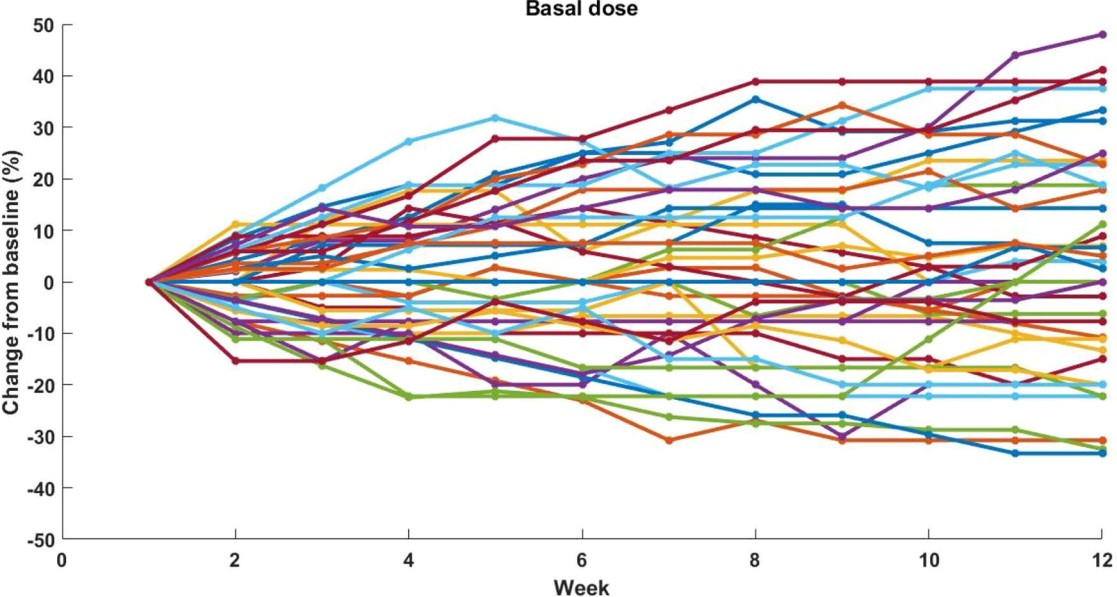

**Fig. 3 | Relative algorithm-made changes in basal insulin doses over 12 weeks.** Relative percent changes in basal insulin doses from baseline over 12 weeks in experimental participants (*n* = 42). Each line color represents an individual participant.

insulin parameters (Supplementary Table 3). However, the algorithm made substantially larger individual adjustments (both increases and decreases) in the insulin dose parameters among experimental participants (Figs. 3–4), compared to the minimal changes observed, as per standard of care, in the control group.

Post-hoc analyses revealed that 15 (36%) and 9 (21%) participants in the experimental group experienced >20% and >30% absolute changes in basal doses, respectively, compared to only 2 (4.8%) and 2 (4.8%) in the control group. Similarly, 19 (45%) and 13 (31%) of experimental participants underwent >20% and >30% absolute changes in prandial bolus doses, respectively, versus only 3 (7.1%) and 1 (2.4%) in the control group (Supplementary Table 4). Notably, while the algorithm made large individual changes to insulin dose parameters, the direction of change (increase vs. decrease) was balanced overall, highlighting the importance of personalized dosing. Specifically, 12 (29%) participants in the experimental group had algorithm-guided increases in both basal and prandial insulin parameters in the final week compared to baseline, 11 (26%) had decreases in both, 9 (21%) had increased basal but decreased prandial insulin parameters, and 10 (24%) had decreased basal but increased prandial parameters (Supplementary Table 5).

### Exploratory outcomes

As part of a prespecified exploratory sub-study, 11 participants in each group completed a modified version of the mHealth App Usability Questionnaire. There were no between-group differences for any of the questionnaire items. Both groups scored very high in all items, suggesting a favorable experience with the standalone iBolus app. (Supplementary Fig. 4).

A total of 24 one-on-one, semi-structured exit interviews were conducted with a subgroup of trial participants (12 experimental and 12 control). Three broad themes emerged from the analysis: 1) enhanced glycemia due to personalized dose recommendations (algorithm-related), 2) advantage of digital solutions over traditional standard of care (related to the standalone iBolus app), and 3) desire for advanced MDI technologies in practice (future directions). From those, 8 sub-themes were identified, with selected ones reported below. The complete list of themes and sub-themes are listed in Supplementary Table 6. The interview topic guide is provided in Supplementary Table 7. Selected quotations from participants are included to illustrate lived experiences in their own words[20].

One prominent sub-theme pertaining to the algorithm was streamlined access to timely decision support. All 12 (100%)

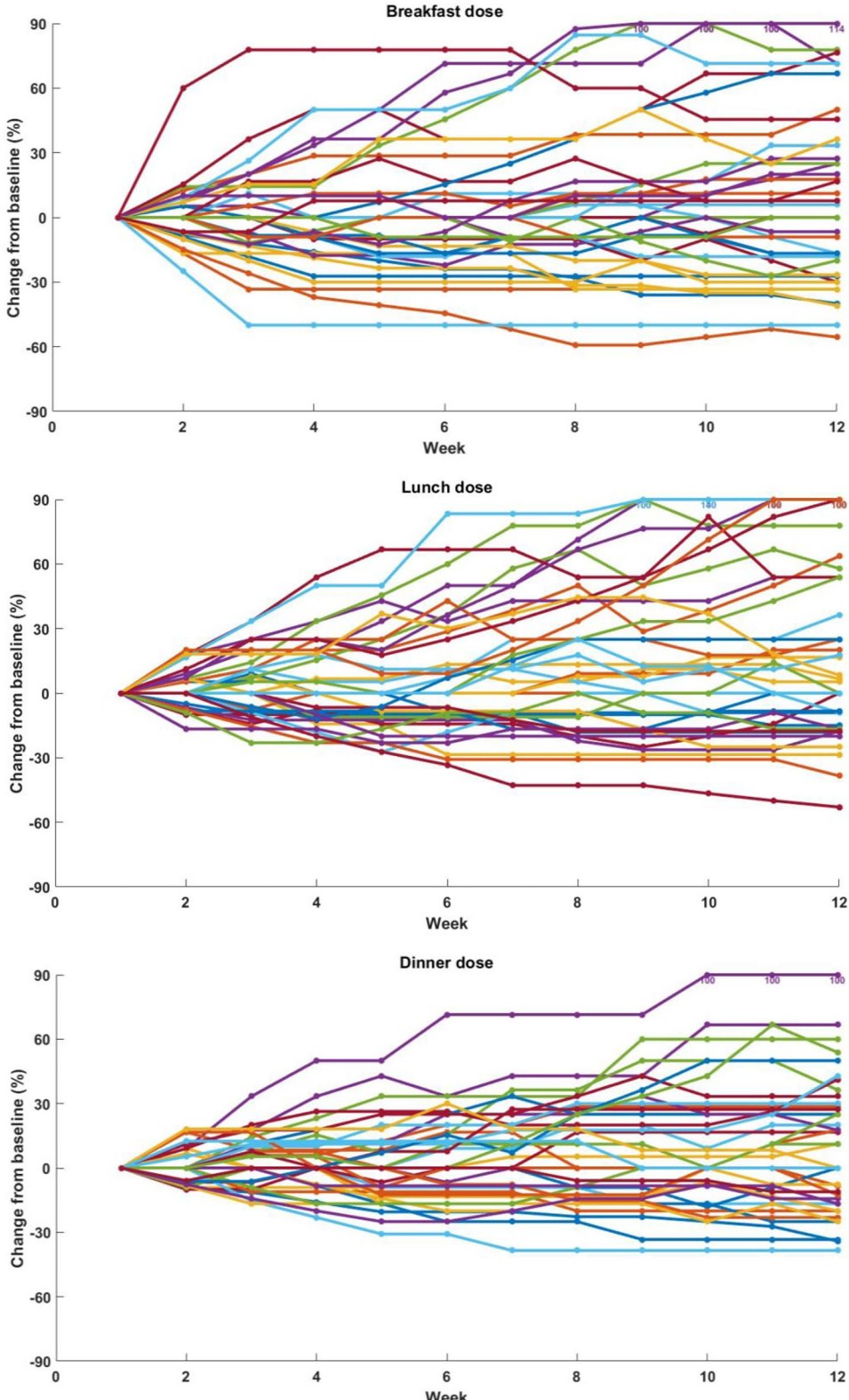

**Fig. 4 | Relative algorithm-made changes in prandial insulin parameters over 12 weeks.** Relative algorithm-made changes in prandial bolus parameters from baseline over 12 weeks in experimental participants ($n = 42$), shown separately for breakfast (top panel), lunch (middle panel), and dinner (bottom panel). Each line color represents an individual participant.

participants in the experimental subgroup expressed a sense of gratitude for receiving frequent insulin adjustments compared to long wait times in traditional settings:

The 6-month patient visit in the hospital, um, to let that go for such a long period of time could have negative consequences to one's health. This [algorithm], at least gives, you know, a

**Table 4 | Adverse events**

| Outcome | Experimental arm (n = 42) | Control arm (n = 42) |
|---|---|---|
| No. of severe hypoglycemia events | 0 | 0 |
| No. of diabetic ketoacidosis events | 0 | 0 |
| *Total no. of adverse events* | **22** | **4** |
| Acute relapse of neuro-pathic pain | 1 | 0 |
| Prolonged hyperglycemia (due to spoiled insulin)* | 1 | 0 |
| Positive COVID-19 infection | 3 | 1 |
| Other upper respiratory infection | 9 | 2 |
| Bacterial throat infection | 1 | 0 |
| Allergic reaction to insect bite | 1 | 0 |
| Bone fracture | 3 | 1 |
| Diarrhea (led to hospitalization) | 1 | 0 |
| Acute relapse of vertigo | 1 | 0 |
| Vomiting | 1 | 0 |

*Participant did not test for ketones and only notified the research team of the hyperglycemia upon resolution with insulin replacement.

guidance immediately in real-time, and again, it does give the patient the option to override. *(P116, experimental)*

Another algorithm-related sub-theme that emerged from the data was appreciation for the conservative (gradual and incremental) nature of dose adjustments, expressed by half (50%) of the experimental subgroup:

It's not a crazy jump week over week when there are changes…I was happy to see that…sometimes there's only one adjustment… it's gradual, it slowly adjusts to the required dosage. *(P126, experimental)*

An additional sub-theme that arose during analysis was trust in technology, whereby 10 (83%) experimental participants expressed a strong feeling of trust with the algorithm:

I had complete confidence in the recommendations, it was going really well, you could really see that there was a good change. *(P115, experimental)*

Several sub-themes also emerged from the standalone bolus calculator. One key sub-theme that surfaced was an enhanced dosing regimen from real-time insulin-on-board tracking and insulin dose history. Overall, nine (75%) control and six (50%) experimental participants stated that the real-time active insulin tracking was a critical feature in facilitating their daily insulin management:

What I liked the most was being able to know how many units of insulin were onboarding, so you know, not to like stack my insulin doses, that, by far was my favorite thing of the entire app. *(P025, control)*

Another sub-theme that arose from 7 (29%) of the 24 interviewees was experiential learning, whereby participants stated that they used the app experientially to enhance their diabetes management:

I was using the app in ways where if I had a low blood sugar, I would kind of see where I can get some free carbs…putting in 15

or putting in 20 grams and seeing how high [in carbs] I can go before it tells me I need one unit [of insulin]. *(P124, experimental)*

During the study, concurrent real-time CGM use was balanced between the groups, with eight participants in each arm wearing their personal Dexcom G6 alongside the intermittently scanned study sensor. Of those, three experimental and five control participants granted sharing access to their personal sensor data for a prespecified exploratory sub-analysis. We obtained $47 \pm 29$ days of overlapping sensor data and observed discrepancies between the sensor types. The intermittently scanned CGM had more readings in the hypoglycemic ranges and less readings in the hyperglycemic ranges compared to real-time CGM, all $p < 0.05$ (Supplementary Table 8).

**Safety**

There were no episodes of severe hypoglycemia or diabetes ketoacidosis throughout the study (Table 4). There were three serious adverse events in the experimental group, none of which were deemed related to the study. One involved hip surgery following a ski accident, another involved overnight hospital admission due to gastrointestinal issues, and the other was a bone fracture caused by an accidental misstep at home that required a medical intervention and evaluation for surgery. There was also one serious adverse event in the control group, comprising a bone fracture caused by an accidental stumble at home that required a medical intervention and evaluation for surgery. Overall, there were more adverse events in the experimental group ($n = 22$) compared to control ($n = 4$), with upper respiratory illness covering the majority of events ($n = 12$; Table 4).

## Discussion

We conducted a randomized controlled trial comparing a DSS for weekly insulin adjustments compared to a standalone app with a bolus calculator in adults with type 1 diabetes and suboptimal glycemic control. The DSS reduced mean HbA1c significantly compared to the standalone bolus calculator. Moreover, it significantly increased the proportion of participants with HbA1c improvements ≥0.5%, nearly doubling the rate compared to the non-adaptive app, and substantially decreased the proportion of those with HbA1c worsening ≥0.5% by -tenfold. These improvements in glycemic outcomes were preceded by large and individualized changes to basal and prandial insulin parameters by the DSS. Other randomized trials have assessed DSSs in adults with type 1 diabetes on MDI therapy[17–19]; however, our trial demonstrated a meaningful improvement in glycemia, in the form of -0.4% control-adjusted HbA1c reduction.

In our study, HbA1c reductions were not accompanied by improved CGM outcomes, which contradicts studies of closed-loop insulin pumps[21,22]. This might be due to the inherent glycemic variability in MDI users caused by the low frequency with which they dose basal insulin (once per day). Our algorithm analyzes glucose data from the past seven days before recommending insulin changes, which subsequently leads to fluctuations between weekly adjustments as a result of less control. In contrast, closed-loop insulin pumps adjust insulin every 5-10 minutes based on real-time CGM readings, resulting in more control and thus less daily glycemic variability. It is, therefore, possible that our sample size was not large enough to detect statistically significant differences in CGM outcomes for an MDI population. Moreover, we used first-generation intermittently scanned sensors, which have been reported to be less accurate than real-time sensors[23], and that may have further increased glycemic variability as measured by the sensor.

Unlike our study, several prior randomized controlled trials tested other DSSs but did not show improvements in glucose control[17–19]. It is difficult to speculate the reasons for the differing outcomes across studies, particularly given the limited publicly available details about the algorithms and mobile apps used in prior trials. However, several

factors may account for the positive findings observed in this trial, such as differences in algorithm logic, app interface, study design, and participant characteristics, among others. A key distinguishing feature of our study is the inclusion of participants with suboptimal baseline HbA1c levels (≥7.5%), which likely influenced the potential for meaningful improvement.

The percentage of time below 3.9 mmol/L observed in our entire study cohort was higher than the recommended target of <4%[24]. However, these higher values were observed in both groups and thus does not indicate a safety signal with our DSS. This is further supported by the absence of increased perceived hypoglycemia in patient-reported outcomes. Moreover, the first-generation intermittently scanned sensor has been reported to be less accurate in the hypoglycemic range[23]. Our exploratory sensor comparison with real-time CGM showed that the intermittently scanned CGM had more readings in the hypoglycemic range, suggesting that these sensors may have overestimated hypoglycemia in our study. Others have also reported large differences between sensor types[25].

Our study not only demonstrated glycemic benefit with an adaptive algorithm for MDI therapy but also incorporated exit interviews, capturing patient perspectives with the study software. There was an overarching positive reception of the McGill DSS, both as a whole and for the algorithm itself, underscoring its clinical relevance. The participants generally found the algorithm's conservative approach comforting and helpful, which fostered a sense of confidence and perceived safety. This aligns with the low number of algorithm recommendation overrides, the statistically and clinically meaningful HbA1c improvements from baseline, and the absence of reported safety concerns. While trust in the system is critical for wider adoption, these findings reflect the views of sub-study participants and are not intended to represent the broader population. This is consistent with the aims of qualitative research, which focus on depth of experience over generalizability. Together, these mixed-methods findings suggest that the algorithm's clinical benefits may also enhance patient experiences, further supporting the potential adoption of the McGill DSS in practice.

A crossover study design would have offered ~fourfold greater statistical power than a parallel design. However, it was not selected due to the high risk of carry-over effects. Significant insulin dose adjustments recommended by the algorithm could not be ethically reversed, and participants could have modified their dosing in the second period based on prior experience, thereby compromising validity.

Strengths of our study include the use of a randomized controlled design and the inclusion of participants with high baseline HbA1c, which is representative of the general MDI population. Furthermore, participants entered the study using their own long-acting and rapid-acting insulins and insulin pens (with various dose increments), without being switched to a uniform insulin type or pen, which further increased the generalizability of our findings. Another strength is the diversity of our study cohort, with one-third of participants identifying as non-White and slightly more than one-third following a fixed-dose meal strategy. Finally, we stratified participants according to previous sensor use to ensure balance between groups, given that glycemic improvements can be obtained from initiating CGM[26–28].

Our study has several limitations. One limitation was the absence of a pre-randomization run-in period to assess and compare baseline CGM metrics with the last few weeks of the study. The control group also lacked baseline optimization; although, neither group received insulin titrations at baseline. Additionally, control participants were permitted to have their parameters modified by their health care providers without restrictions per their usual standard of care. Notably, despite the high average baseline HbA1c, the study cohort may reflect a more motivated population given their willingness to participate in the study as opposed to those who

declined participation. However, recruitment occurred during the COVID-19 pandemic, which may have contributed to the higher-than-expected number of individuals who declined participation ($n = 37$) or were lost to follow-up after expressing initial interest ($n = 25$). Another limitation was that the DSS did not optimize the insulin sensitivity factor; however, participants could manually adjust this parameter during the intervention, which was done by seven participants. Finally, insulin dosing details were recorded by users through the app, making them subject to reporting error, rather than being automatically captured by a smart pen or cap device. However, all participants received rigorous baseline training and understood the importance of accurate and timely reporting, with additional support by automated compliance reminders in the app.

In conclusion, our DSS significantly improved HbA1c in adults with type 1 diabetes and suboptimal glucose control on MDI therapy.

## Methods
### Trial design
This was a 12-week, multicenter, open-label, parallel, randomized controlled superiority trial conducted in an outpatient setting. Neither participants nor study staff were blinded to group assignment due to the nature of the intervention. Control participants used the non-adaptive app with their prescribed insulin parameters, whereas experimental participants used the adaptive app, which incorporated weekly updates with algorithm-made adjustments to basal and prandial insulin parameters. The protocol was approved by the McGill University Health Center Research Ethics Board and by Health Canada, which granted Investigational Testing Authorization to conduct the trial with the McGill DSS (regulated as a class II investigational medical device). The study protocol was registered on ClinicalTrials.gov (NCT04123054) prior to study initiation.

The study was conducted primarily at the McGill University Health Center, Royal Victoria Hospital (Montreal, Canada). It was later expanded to include the Jewish General Hospital (Montreal, Canada) as a referral site. Following authorization from the Convenance of the Integrated Health and Social Services University Network for West-Central Montreal (CIUSSS West-Central Montreal), interested participants from the Jewish General Hospital were referred to the study team for screening, consent, and all study procedures. Written informed consent was obtained from all participants prior to commencement.

The objective was to compare the effectiveness of the McGill DSS, which provides weekly adaptations to basal and prandial bolus parameters, to a non-adaptive bolus calculator. All participants were supplied with first-generation Freestyle Libre sensors. Participants were allowed to continue their personal real-time CGM, if applicable, alongside the study CGM.

### Participants
Inclusion criteria included males and females ≥18 years of age with a clinical diagnosis of type 1 diabetes for at least 12 months, using multiple daily injections, and with a baseline HbA1c value ≥ 7.5% (measured within ± seven days of screening). Exclusion criteria comprised any of the following: inability or unwillingness to comply with the study protocol, use of intermediate-acting insulin, use of more than one daily long-acting insulin injection and unwillingness to switch to once-daily dosing, use of non-insulin antihyperglycemic agents within one month, pregnancy, severe hypoglycemia or diabetic ketoacidosis within one month, clinically significant nephropathy, neuropathy, or retinopathy (per the investigator's judgment), conditions affecting HbA1c accuracy (including clinically significant anemia, hemoglobinopathy, or recent blood transfusion), acute macrovascular event within six months, or other serious medical illnesses likely to interfere with study participation or completion (per the investigator's judgment). Stopping criteria included protocol non-compliance,

pregnancy, or investigator judgment that continued participation was not in the participant's best interest.

Most participants were recruited at the endocrinology clinic of the McGill University Health Center. If a participant had an eligible HbA1c value collected within one week of the screening visit, it was used as the baseline measurement; otherwise, a new baseline HbA1c measurement was taken during the screening visit.

### Randomization

We utilized a computer-generated permuted-block 1:1 randomization, stratified according to regular sensor use (≥3 months) or new sensor use, with different block sizes of 10 and 4, respectively, for each stratum. A member of the research team who was not involved in clinical aspects of the study prepared a computer-generated randomization sequence and embedded it into sequentially numbered, digitally sealed envelopes, which were opened only after enrollment was confirmed. The allocation sequence was concealed from the investigators and the study coordinator responsible for enrolling participants, who had no access until the point of assignment.

### Procedures and interventions

Following randomization, participants received refresher training on diabetes management that included a review on pharmacokinetic action of insulin analogs, insulin storage and care, injection techniques, carbohydrate counting (if applicable), and hypoglycemia and hyperglycemia management.

Study LibreLink app accounts were created for each participant using study coded credentials, followed by training on the insertion, use, and disposal of the study sensors. All participants were asked to scan their Freestyle libre sensor at least five times per day (upon waking, prior to meals, and before bedtime) and input the scanned values in the iBolus app upon calculating a meal or correction bolus dose. Sensor readings were automatically transferred to our LibreView Professional account and downloaded weekly into a secure drive and database.

Initial iBolus app programming involved enabling the carbohydrate counting setting or keeping it disabled for fixed-dose users, along with inputting their personal dosing parameters from their existing treatment regimen, including basal dose, glucose target, insulin sensitivity factor, active insulin time, basal and bolus pen increments, and carbohydrate ratios or fixed doses for each meal type. The app was designed to lock the selected meal strategy (carbohydrate counting or fixed-dose) at initiation, preventing participants from switching regimens after starting the study. Participants in both groups received training on the iBolus app and were instructed to use it daily and accurately for timely basal logging and real-time meal and correction bolus calculations. The control group used the non-adaptive iBolus app, which relied on their personal programmed parameters, and the experimental group used the adaptive version of the app.

Adverse events were systematically assessed during the follow-up call and end-of-study visit. Additionally, participants were instructed to self-report any adverse events, related or unrelated to type 1 diabetes, between visits by direct contact with the study coordinator.

### The McGill decision support system

**iBolus app.** The iBolus app was developed with the aim of facilitating bolus calculations for MDI therapy while simplifying remote data collection in the study. The app was designed with a focus on a simplified user interface with icon-based features while minimizing text (Supplementary Fig. 5).

All participants had downloaded the iBolus app for daily use. The app is equipped with a built-in calculator that computes meal and correction bolus doses in real-time based on programmed parameters (meal-specific carbohydrate ratio or fixed-dose, a pre-set glucose

target, and insulin sensitivity factor) and the current glucose level and corresponding trend, while accounting for active insulin-on-board from the previous rapid-acting insulin dose. The app also included basal dose logging, with an optional notification that could be enabled as a daily reminder at the usual administration time. In the event of an error, users had a 30-minute window to delete any entry. Automatic alerts were triggered if three consecutive basal logs were missed or if app use was insufficient, defined as three or fewer entries within 36 h. These alerts served to promote adherence by reminding participants to engage with the app.

Upon requesting a prandial bolus dose, the meal type (breakfast, lunch, dinner, or bedtime) was set by default depending on the time of day. Then, participants were prompted to enter their sensor glucose value and select the corresponding trend arrow, and enter the amount of carbohydrates, if applicable. The calculated dose was subsequently displayed along with the calculation details. Upon confirming or overriding a calculated dose, the app automatically updated the insulin-on-board and displayed it in real time on the home screen. When overriding a dose, participants could optionally select a reason (dose is higher than usual, exercise, sick, stress, menstrual cycle, alcohol). These reasons were stored as notes and did not affect the bolus dose calculation. For the correction bolus, participants were prompted to enter their sensor glucose value and trend arrow to calculate the dose, which they could also accept or override.

The non-adaptive iBolus app included the aforementioned features, which relied on programmed personal parameters that participants could manually change at any time. In contrast, the adaptive version (i.e., integrating the algorithm to form the McGill DSS) entailed weekly notifications that displayed the previous week's basal and prandial bolus parameters along with the new optimized parameters determined by the algorithm. Upon acknowledgement, the new parameters automatically updated in the app settings and were applied the following week.

App data were automatically transferred to a secure cloud server upon internet connection. Notifications for missing data and the weekly parameter notifications also required internet connection; however, all other app features functioned locally. The iBolus app was installed on the participant's personal phone, using study credentials, if compatible, or preloaded on a study-issued phone.

**Optimization algorithm.** Our algorithm employed a model-based Bayesian approach to estimate insulin dosing parameters (once-daily basal dose and fixed prandial bolus doses or carbohydrate ratios)[29]. Once a week, the algorithm was executed on the cloud (MATLAB R2018b running on a Google Cloud Virtual Machine) for all active experimental participants. The algorithm utilized the previous week's sensor data, obtained from the Freestyle LibreView Pro study platform, and insulin and meal data obtained from the iBolus app (which was securely transferred to a Google Firebase server) as input to fit a glucoregulatory model using a Bayesian approach to estimate the therapy parameters that would have resulted in optimal control in the previous week. In addition to the model-based recommendations, the algorithm had an additional safety rule to make the recommended insulin amount less aggressive when postprandial glucose levels were trending low. The recommendations for the following week were a mixture of these optimal parameters and the parameters used in the previous week depending on the ability of the model to explain the data and the statistical confidence in the optimal parameters' estimates. The algorithm recommendations were subsequently pushed to the participants in the form of an app notification, requiring acknowledgement to automatically populate their app settings.

The algorithm was executed weekly; however, all previous weeks were implicitly included in the algorithm recommendations. Each week, the algorithm used the Bayesian's prior probabilistic

distributions to combine the previous week's data with the most recent therapy parameters (which were learned from former weeks) to generate recommendations for the following week. This iterative process ensures that lessons from previous weeks are carried to future recommendations.

The app and the algorithm were accompanied by an online platform that was accessed by the research team only for data visualization. If there was more than a 30% cumulative change in any therapy parameter from baseline or a previously approved parameter, then approval from a member of the study's clinical team was required in the platform before the new parameters were pushed to the participant's app.

### Patient-reported outcomes

As part of a prespecified exploratory sub-study, participants were enrolled based on convenience sampling, wherein any new participant enrolled in the main study was given the opportunity to participate in this sub-study with a target range of 20-25 individuals. The patient-reported outcomes were based on a modified version of the mHealth App Usability Questionnaire, administered at the end-of-study visit, to gauge the usability of the standalone iBolus app.

The mHealth App Usability Questionnaire is a validated survey for a standalone mobile health application. This instrument is composed of 18 items, broken down into three factors: 1) ease of use, 2) interface and satisfaction, and 3) usefulness[30]. All items are scored on a 7-point Likert scale ranging from 1 (disagree) to 7 (agree). One question from the ease-of-use factor and another from the usefulness factor were removed due to their irrelevance to the iBolus app.

### Interview outcomes

Qualitative assessments were conducted to capture the lived experiences of a participant subgroup, enriching the quantitative findings with insights into the perceived utility of the technology. The subgroup of participants who completed questionnaires also took part in semi-structured one-on-one exit interviews led by a topic guide (Supplementary Table 7). The interviews were audio recorded, transcribed verbatim, and then underwent a thematic analysis by two coders (A.K. and A.H.). The codes were developed inductively without predefined codes, albeit inherently through a deductive lens due to prior experience, knowledge, and preconceived notions. This approach was taken to better understand participants' perspectives and to assess the usability of the study software.

### Primary and secondary endpoints

The primary endpoint was the change from baseline in HbA1c at end-of-study. Secondary endpoints included the percentage of time for which sensor glucose levels were in the following ranges: between 3.9–10 mmol/L, between 3.9–7.8 mmol/L, <3.9 mmol/L, <3 mmol/L, >7.8 mmol/L, >10 mmol/L, >13.9 mmol/L, and >16.7 mmol/L, as well as standard deviation, mean sensor glucose, and total insulin dose. Data from CGM metrics were calculated for three periods: 1) overnight (23:00–7:00), daytime (7:00–23:00), and overall (24-h).

### Statistical methods and analyses

The study was designed to test the primary hypothesis. Based on prior literature[27], we assumed a standard deviation of 0.8% for HbA1c and an estimated between-group difference of 0.5%. Under these assumptions, a total sample size of 84 participants (42 per arm) was calculated to provide 80% power to detect a statistically significant difference at a two-sided $\alpha$ of 0.05.

The primary endpoint was compared using a linear mixed model adjusting for baseline HbA1c. Hypothesis testing of between-group differences in other continuous outcomes was performed using the independent two-sample t-test for normally distributed data or the Wilcoxon rank-sum (Mann-Whitney) test for non-normally distributed

data. The Shapiro-Wilk test was used to assess normality. Data were reported as mean (standard deviation) for normally distributed variables and median [interquartile range] for non-normally distributed variables. The primary analysis followed a modified intention-to-treat approach, including all randomized participants with a final HbA1c measurement after at least 10 weeks of intervention. Prespecified secondary analyses were conducted using all available data from randomized participants. Subgroup analyses were carried out as defined in the protocol, while post-hoc analyses were performed to further explore associations and outcome patterns.

The statistical analyses for the quantitative data were performed using R v12.1 and MATLAB 2020. All between-group comparisons were two-tailed, with p values < 0.05 considered statistically significant. No corrections for multiplicity were applied to secondary outcomes.

For the qualitative sub-study, we aimed to include a range of 20–25 participants to assess the impact of the study software under free-living conditions. We adopted a qualitative descriptive methodology to explore and describe participants' experiences and views with minimal inference from the data[31]. A thematic analysis was employed to identify recurring patterns, which were grouped into themes to summarize the data. The NVivo 12 software was used to manage, store, and analyse the qualitative data.

### Reporting summary

Further information on research design is available in the Nature Portfolio Reporting Summary linked to this article.

## Data availability

The study protocol is provided in the Supplementary Information. Raw data cannot be made publicly available due to restrictions in the informed consent form. However, deidentified individual participant data, including baseline characteristics and outcome measures (HbA1c, CGM and insulin metrics, and survey responses) are available upon request. Requests should be directed to the corresponding author by email. Data will be shared at no cost for non-commercial research purposes, subject to approval by the Research Ethics Board of the McGill University Health Center. Following approval, data will be transferred securely within three months. Data will become available three months after publication and will remain available for five years. Source data are provided with this paper.

## Code availability

The code used for data processing is available at https://github.com/McGillDiabetesLab/MDI2020-analysis. The algorithm code is a proprietary intellectual property and therefore cannot be made publicly available.

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

## Acknowledgements

We wish to thank the study participants. Internally, we thank the engineering support team: Alexis Giguère-Joannette (mobile app developer), Joanna Rutkowski (technical support in the study), and Robert Kearney (algorithm co-development). We also wish to thank Nikita Gouchie-Provencher for her delegated medical assistance. Finally, we wish to thank Dr. Stephanie Michaud, Dr. Sara Meltzer, Dr. Melissa-Rosina Pasqua, and Dr. Tricia Peters for recruitment support. This study was funded by the Canadian Institute of Health Research (Grants # 950-231305 and 356406 held by Ahmad Haidar and Robert Kearney). The Canadian Institute of Health Research had no role in study design, data collection and analysis, or manuscript writing.

## Author contributions

A.K., A.H. and A.E.F. designed the main study and mobile app; A.K. and A.H. designed the sub-studies. A.K. developed the study documents, led the trial operations (including recruitment, study procedures, and data management), and conducted the qualitative sub-study. A.E.F. and A.H. developed the algorithm. M.A.T. provided medical oversight as the principal investigator. N.G., M.A.T., and L.L. conducted the medical assessments, and N.G. and J-F.Y. supported recruitment. A.J. processed the data. A.K. and A.H. performed the statistical and qualitative analyses and interpreted the results. A.K. drafted and finalized the manuscript with A.H.'s feedback. A.H. is the guarantor of this work; as such, he had full access to the data and takes responsibility for the integrity of the data analysis. All authors reviewed and approved the final version of the manuscript.

## Competing interests

A.H., A.E.F and A.J. own intellectual property in diabetes technology. A.J., A.H., and M.T. received IP acquisition fees from BigFoot; A.H. also from Eli Lilly. A.H. received research support from Eli Lilly, Dexcom, Adocia, Tandem, Bet Bionics and AgaMatrix, and consulting fees from Eli Lilly and Abbot Diabetes. L.L. received research support (AstraZeneca, Merck), consulting fees (Abbott, Dexcom, Insulet, Novo Nordisk), and speaker/travel support (Novo Nordisk). J-F.Y. received research support

(Bayer, Novo Nordisk, Novartis, Sanofi), consulting fees (Abbott, Dexcom), and speaker honoraria (Eli Lilly, Novo Nordisk). M.A.T. received research support (AgaMatrix) and speaker honoraria (Eli Lilly, Novo Nordisk, Boehringer Ingelheim, Janssen, AstraZeneca). All other authors declare no competing interests.

## Additional information

https://doi.org/10.1038/s41467-025-63671-0).

