## [Transparent Peer Review file · Nature Communications]

A Bayesian decision support system for automated insulin doses in adults with type 1 diabetes on multiple daily injections: a randomized controlled trial

Corresponding Author: Dr Ahmad Haidar

Version 0:

Reviewer comments:

Reviewer #1

(Remarks to the Author)

This article describes the results of 12 week randomized clinical trial of 84 adults with type 1 diabetes with baseline A1c >7.5 who were randomized to a bolus calculator system with or without a decision support system (DSS). The DSS was an algorithm that provided recommendations for adjustments of prandial and long acting insulin doses once weekly. The participants use a flash glucose monitoring system and remained on their usual home insulin without dose tracking pen. In the DSS intervention participants showed reduced A1c from 8.6% to 8.1% versus control (bolus calculator alone) 8.6% to 8.5%, this was the primary outcome. Despite these positive A1c results, there were no statistically significant changes in any of the CGM parameters assessed. Notably the percent time <3.9% was above 4% in both intervention and control arms at 5.9% and 4.3% respectively which was not statistically significantly different between the arms. A few participants in each arm wore a real-time CGM concurrently to the flash CGM, the real-time CGM showed substantially lower time <3.9 which the authors reasonably attributed to better accuracy of the real-time CGM in these lower ranges. They also assessed patient reported outcomes in semi structured interviews in a subset of the participants with favorable results.

This is an interesting and important study in that it is the first randomized study of this type of DSS that showed a clinically significant improvement in A1c in the intervention group. This type of system is greatly needed for real world use in people with T1D and in future work in T2DM as well. The study has some limitations. The lack of direct dose tracking means that we cannot be certain if the recommendations of the DSS system were followed which may have blunted the full effect of the DSS. The flash CGM system used is now out of date and it appears unreliable in the hypoglycemia range in comparison to the real-time CGM. The lack of concurrence between unchanged CGM outcomes and improved A1c outcomes is a bit perplexing although the authors provide some reasonable considerations for these outcomes.

1. It would be useful to speak to how the DSS algorithm handles hypoglycemia. In "A Model-Based Insulin Dose Optimization Algorithm for People With Type 1 Diabetes on Multiple Daily Injections Therapy" it states that if hypoglycemia occurs after a meal the system would not recommend a more aggressive carb ratio. Were the results of their study impacted by the inaccuracy of the flash CGM system resulting in what appears to be falsely high time in hypoglycemia?
2. Any subgroup analysis of those who use the app more frequently vs less frequently (i.e. is there a dose effect of the iBolus DSS?)
3. The Supplementary table 1 shows only about 0.5 correction boluses per day in each arm which seems quite low. Does this include only corrections given independently of meals?
4. Supplementary Table 5 Daytime % time 3.9-7.8 p-value reads as 10.48. Perhaps this is a typo.
5. Consider addition limitation to discussion: while baseline A1c was high indicating they enrolled a population somewhat similar to the general population of T1D MDI users, however it is notable that 179 participants had to be screened to enroll 84. 25 people were lost to follow up, so the population in this study represents a more engaged group as also noted by ~3 bolus calculator uses per day.
6. To further investigate the author's hypothesis that the lack of improvement in CGM parameters may be due to inherent glycemic variability and underpowered sample size, consider a similar analysis to Figure 3 for AGP glycemic metrics, such as improvement in TIR >5,10,15% or mean glucose improvement 10,20,30mg/dL.

Reviewer #2

(Remarks to the Author)

Title: A Novel Bayesian Decision Support System for Automated Insulin Doses in Adults with Type 1 Diabetes on Multiple Daily Injections: A Randomized Controlled Trial

Overview:

This is a review of a trial using a DSS for insulin. The algorithm itself is detailed in a previous paper (Fathi, Kearney, Palisaitis, Boulet, Haidar; 2021) and is therefore not part of the review of this paper (the algorithm is taken as a given).

Main points:

One of the key findings (in the abstract and discussion) is the doubling of proportions achieving several threshold improvements in HbA1c. I do not believe those doubling proportions are statistically tested? If they are, those findings (and the uncertainty around those findings) is not clearly presented. Please formally test these doubling proportions, or if already done add the confidence intervals into the main text.

The discussion considers why only HbA1c changes but not CGM are found to be significant. However, it does not relate back to the three studies mentioned in the introduction. What about this algorithm differs from those interventions? Since none of those three studies showed any improvements, what is different here? I think this is a key point that readers would be interested in understanding. Please add more details on the differences with the background DSSs, and some discussion of what aspects this algorithm seems to be handling better for patients (e.g. is it the algorithm itself, or could it just be a better presentation leading to better patient adherence?).

With regard to the trial design. The background in the introduction include three previous studies, two of which are cross-over designs (p5 l78). The current paper gives no discussion on why such an approach was not taken here. Further, the trial was only powered at 80% to detect a change in HbA1c between the two groups. As mentioned above, the paper reports on changes in proportions across various threshold increases (and decreases) that were not directly powered for; neither was CGM part of the sample size consideration (despite being another primary outcome). It is unclear whether CGM should be a key aspect of the paper (the discussion admits the trial is likely under powered for CGM, p9 l238). Please add further discussion on the trial design itself.

The paper combines the quantitative and qualitative findings of the trial. The qualitative aspect is only mentioned in the discussion within the limitations regarding manually adjusting doses. I feel the qualitative findings are not given enough reflection within the discussion. It is not clear if they will be explore in another paper in-depth; please add more reflection on the qualitative findings in the discussion (hopefully linking to the points raised in the above paragraphs).

Minor points:

p5 l123: Was the "almost doubles" tested statistically? It is unclear whether these changes in proportions are robust. Add confidence intervals on these ratio tests.

Figure 3: There should be uncertainty intervals on the barcharts to support the "almost doubles" statements throughout the paper.

p5 l126: The language changes to "substantially" rather than statistically. The reader is left without any definite answer to

whether those differences are reliable. (Continuation of the above points).

p7 I181: The Qualitative section is very brief, it was unclear to what extent this was to support the main paper. Patient acceptability and implementation.

p7 I203: I find the 10/12 (83%) experimental interview participants indicating a trust in the algorithm to be a key line in the wider adoption of this algorithm. There is no discussion of this point, nor a consideration of whether this might be over estimated in a sample of participants willing to engage in trials. Linked to this is the option to over-ride the app.

p7 I195: The article does not report how often participants in the experimental arm overrode the algorithm. Unclear if this is a major aspect, but it is called out specifically so the reader is left to wonder (especially with respect to the above point about trusting the algorithm).

p10 I217: The final sentence of the discussion seems, a bit like a throw away comment. Implementation of a DSS in pediatrics is an entirely different area, with issues of smart phone access (for adolescents), engagement with technology, and managing their own care. I see no reason for these seven words, the previous sentence clearly limits your findings to adults so please remove this last sentence.

Methodology: As a minor point, I find it interesting that a Bayesian methodology is used for the algorithm, but all the analyses are under a frequentist paradigm. Nothing wrong with this, but given the overlap in authors of the algorithm paper and this paper.

Reviewer: Dr Simon R White, University of Cambridge

Reviewer #3

(Remarks to the Author)

Thank you for the opportunity to review this mixed methods paper – describing an open label RCT assessing whether a novel decision support algorithm can improve HbA1c in people with type 1 diabetes who manage with multiple daily injections. The manuscript presents quantitative data for metabolic parameters, and qualitative treatment satisfaction data.

I found the study relevant and novel, and of interest to the academic diabetes community. The study achieved its primary outcome (HbA1c reduction), though I offer a few suggestions to support the validity of the findings. My comments are below:

Results

- Hba1c outcomes – suggest that for completeness within the text, that the authors also report the 95% confidence intervals or standard deviation of the within group differences (including in the abstract). I acknowledge that this is reported in the tables and figures.
- Are there statistics available for the statements regarding the increased proportion of participants with improvements in HbA1c by 0.5%, 1.0% and 1.5%?
- Could the authors confirm that there was balance between groups for participants wearing their own Dexcom G6 (ie. concurrent CGM use, not just previous CGM use)? Participants had to manually enter their mealtime glucose level, and there was mention of a glucose reading discrepancy between the libre and Dexcom sensors (libre reading lower). Balance between groups becomes relevant where possibly, participants may have chosen to input their glucose level from the Dexcom rather than Libre, thereby affecting the insulin dose recommendation from the app.

Methods

- There are 2 figures both labelled supplementary figure 5. In the methods, fig 5 is reference, though I assume is meant to refer to Fig 6.
- Were the adaptive and non-adaptive groups given the same glucose target in the app? This detail may have been missed in my reading, or otherwise could be made more obvious
- Endpoints – does 'insulin delivery' mean insulin dose?

Discussion

- Other RCTs assessing decision support systems for MDI users do not report improvements in glycemia. The discussion could be strengthened with some comment about possible reasons for the favourable result – differences in the McGill

algorithm, or differences in the patient population compared to other studies?

Other minor comments:

- the supplementary tables are out of order, relative to where they are referenced in the text
- Supp table 2 – could prandial insulin also be reported as Units/day, since basal is reported as units per day. It would be helpful to see this side by side.
- Could the statistics used to determine the p-value please be included as a footnote under each table.

Version 1:

Reviewer comments:

Reviewer #1

(Remarks to the Author)

The authors have adequately addressed all reviewers comments.

Reviewer #2

(Remarks to the Author)

Title: A Novel Bayesian Decision Support System for Automated Insulin Doses in Adults with Type 1 Diabetes on Multiple Daily Injections: A Randomized Controlled Trial

Revised manuscript:

The authors have addressed the concerns raised in my review.

Reviewer: Dr Simon R White, University of Cambridge

Reviewer #3

(Remarks to the Author)

Sending thanks to the authors for their careful revisions. I am satisfied with their responses to my comments.

REVIEWER COMMENTS

Reviewer #1

1. It would be useful to speak to how the DSS algorithm handles hypoglycemia. In “A Model-Based Insulin Dose Optimization Algorithm for People With Type 1 Diabetes on Multiple Daily Injections Therapy” it states that if hypoglycemia occurs after a meal the system would not recommend a more aggressive carb ratio. Were the results of their study impacted by the inaccuracy of the flash CGM system resulting in what appears to be falsely high time in hypoglycemia?

Response: We thank the reviewer for this question. Even if the sensor may not have been very accurate in the hypoglycemia range, we do not believe that this jeopardized the application of the safety rule with postprandial hypoglycemia. First, even if the safety rule were to be triggered following a false hypoglycemia, glucose levels would likely have been trending close to the hypoglycemia range, and thus decreasing prandial insulin would have been the right decision anyway. Second, this rule is an additional safety layer on top of the model-based recommendations, and the model would normally recommend decreasing prandial insulin if postprandial glucose levels were trending low.

Description of the hypoglycemia safety rule is now provided in the revised manuscript (page 16).

2. Any subgroup analysis of those who use the app more frequently vs less frequently (i.e. is there a dose effect of the iBolus DSS?)

Response: We plotted the individual overall app use vs. change in HbA1c (see new Supplementary Fig 3), which shows that HbA1c improvements was not related to greater app usage. This suggests that app usage may have been driven by the number of meals consumed per day, as more frequent app usage does not necessarily translate into improved glucose control. A statement about this was added to the revised manuscript (page 5).

3. *The Supplementary table 1 shows only about 0.5 correction boluses per day in each arm which seems quite low. Does this include only corrections given independently of meals?*

Response: Yes, that is correct. These are the numbers for the correction boluses outside mealtimes. This has been clarified in the revised manuscript (page 4). The purpose of this table is to report the average user interactions with different app features, and thus we are reporting the number of times the participants used the correction bolus feature to deliver corrections outside mealtimes.

4. *Supplementary Table 5 Daytime % time 3.9-7.8 p-value reads as 10.48. Perhaps this is a typo.*

Response: Thank you for pointing out this typo. The correct p value is 0.48. It has been corrected.

5. *Consider addition limitation to discussion: while baseline A1c was high indicating they enrolled a population somewhat similar to the general population of T1D MDI users, however it is notable that 179 participants had to be screened to enroll 84. 25 people were lost to follow up, so the population in this study represents a more engaged group as also noted by ~3 bolus calculator uses per day.*

Response: We thank the reviewer for this comment. An additional statement was added to the limitation paragraph (page 11) that the study cohort might have been a more motivated cohort than the general population given their agreement to participate in the study. Note, however, that the study recruitment happened during the covid pandemic, which may have contributed to the number of individuals who declined to participate (n=37) or lost to follow up after initially showing interest in participating in the study (n=25).

6. *To further investigate the author's hypothesis that the lack of improvement in CGM parameters may be due to inherent glycemic variability and underpowered sample size, consider a similar analysis to Figure 3 for AGP glycemic metrics, such as improvement in TIR >5,10,15% or mean glucose improvement 10,20,30mg/dL.*

Response: We thank the reviewer for the suggestion. We have performed this analysis and found no difference between the two arms when comparing the first and last two and four weeks of those TIR and mean glucose improvements. A statement reporting this was added to the revised manuscript (page 5).

Reviewer #2

Main points:

1. *One of the key findings (in the abstract and discussion) is the doubling of proportions achieving several threshold improvements in HbA1c. I do not believe those doubling proportions are statistically tested? If they are, those findings (and the uncertainty around those findings) is not clearly presented. Please formally test these doubling proportions, or if already done add the confidence intervals into the main text.*

Response: We thank the reviewer for this comment. We have removed Figure 3 and replaced it with Table 3, which compares the proportions between the two arms and includes corresponding p-values and 95% confidence intervals. The only statistically significant between-group differences were observed for the HbA1c degradation threshold of $\geq 0.5\%$ and the HbA1c improvement threshold of $\geq 0.5\%$. Note, however, the study was not powered to detect differences in proportions. Accordingly, we also removed these outcomes from the abstract, as they were not pre-specified primary outcomes while also making room for an addition requested by the third reviewer.

2. The discussion considers why only HbA1c changes but not CGM are found to be significant. However, it does not relate back to the three studies mentioned in the introduction. What about this algorithm differs from those interventions? Since none of those three studies showed any improvements, what is different here? I think this is a key point that readers would be interested in understanding. Please add more details on the differences with the background DSSs, and some discussion of what aspects this algorithm seems to be handling better for patients (e.g. is it the algorithm itself, or could it just be a better presentation leading to better patient adherence?).

Response: It is difficult to speculate about the reasons for different outcomes between studies. The differences in outcomes could be related to differences in the learning algorithms' design, the apps' design, the study designs, or the study populations, among others. This is now stated in the revised manuscript (page 9).

Note also that the details of algorithms and apps' designs used in the other studies are also not fully publicly available, which makes it difficult to compare them to our system.

3. With regard to the trial design. The background in the introduction include three previous studies, two of which are cross-over designs (p5 l78). The current paper gives no discussion on why such an approach was not taken here. Further, the trial was only powered at 80% to detect a change in HbA1c between the two groups. As mentioned above, the paper reports on changes in proportions across various threshold increases (and decreases) that were not directly powered for; neither was CGM part of the sample size consideration (despite being another primary outcome). It is unclear whether CGM should be a key aspect of the paper (the discussion admits the trial is likely under powered for CGM, p9 l238). Please add further discussion on the trial design itself.

Response: Thank you for this comment. The crossover design would have provided us with 4-fold more statistical power. However, we opted against it as we believe the risk of carry-over would have been high. For example, if a participant started with the algorithm intervention, and the algorithm made significant changes to their daily insulin doses (e.g., reduce basal significantly to eliminate night hypoglycemia), then ethically we would have needed to start with these (supposedly better) doses for the second intervention, which would have been a direct carry over effect. Even if we return the participants to their baseline insulin doses after the first algorithm intervention, they might opt to change their doses immediately after the start of their second intervention based on their observation of the first intervention, which is also a direct

carryover effect. The rationale of opting against a crossover design was added to the manuscript (page 10).

Note that the CGM was not a pre-defined primary outcome (only HbA1c was). We do not believe there is no right or wrong choice when it comes to the primary outcome between CGM or HbA1c. If we had chosen CGM time in range as our primary outcome, we would have modified our inclusion criteria to include only those with low (e.g., < 40%) time in range and might have required a larger sample size and longer study.

4. The paper combines the quantitative and qualitative findings of the trial. The qualitative aspect is only mentioned in the discussion within the limitations regarding manually adjusting doses. I feel the qualitative findings are not given enough reflection within the discussion. It is not clear if they will be explore in another paper in-depth; please add more reflection on the qualitative findings in the discussion (hopefully linking to the points raised in the above paragraphs).

Response: Thank you for raising this point. We have added a paragraph in the discussion to discuss the qualitative component of the study (p.10).

Minor points:

1. p5 l123: Was the "almost doubles" tested statistically? It is unclear whether these changes in proportions are robust. Add confidence intervals on these ratio tests.

Response: Yes, we have now reported the p values comparing the proportions between the two arms in the text (page 5). We also included Table 3, which contains the confidence intervals.

2. Figure 3: There should be uncertainty intervals on the barcharts to support the "almost doubles" statements throughout the paper.

Response: We have removed Figure 3 and replaced it with Table 3, which compares the proportions between the two arms and includes corresponding p-values and 95% confidence intervals

3. p5 l126: The language changes to "substantially" rather than statistically. The reader is left without any definite answer to whether those differences are reliable. (Continuation of the above points).

Response: We have added p values and confidence intervals for the differences in proportions in the text and in Table 3.

4. p7 l181: The Qualitative section is very brief, it was unclear to what extent this was to support the main paper. Patient acceptability and implementation.

Response: Thank you for raising this point. Qualitative assessments are common in our field to capture patient's lived experiences and to allow for the comprehensive assessment of the technologies. We have tweaked the qualitative section in the results (pages 7-8), elaborated more

on the qualitative section in the discussion (page 10), and added clarification to the methods (pages 16-17).

5. p7 l203: I find the 10/12 (83%) experimental interview participants indicating a trust in the algorithm to be a key line in the wider adoption of this algorithm. There is no discussion of this point, nor a consideration of whether this might be over estimated in a sample of participants willing to engage in trials. Linked to this is the option to over-ride the app. p7 l195: The article does not report how often participants in the experimental arm overrode the algorithm. Unclear if this is a major aspect, but it is called out specifically so the reader is left to wonder (especially with respect to the above point about trusting the algorithm).

Response: We thank the Reviewer for this comment. We agree that the high trust in the technology would be important for its wider adoption. We also agree that our numbers should be treated with caution as our participants may not be representative of the general population. This was stated in the revised manuscript (page 10).

We had a low number of algorithm recommendations overrides. The algorithm recommendations were overridden 41 times (out of 504 total recommendations). This is now specified in the revised manuscript (page 4).

6. p10 l217: The final sentence of the discussion seems, a bit like a throw away comment. Implementation of a DSS in pediatrics is an entirely different area, with issues of smart phone access (for adolescents), engagement with technology, and managing their own care. I see no reason for these seven words, the previous sentence clearly limits your findings to adults so please remove this last sentence.

Response: Thank you for raising this point. We have removed this sentence.

7. Methodology: As a minor point, I find it interesting that a Bayesian methodology is used for the algorithm, but all the analyses are under a frequentist paradigm. Nothing wrong with this, but given the overlap in authors of the algorithm paper and this paper.

Response: We adopted a Bayesian algorithm to be able to incorporate the previous week insulin parameters as priors for the following week, as the glucose and insulin data were not always rich enough to allow for a strong identification of the insulin parameters. For the clinical trial design, we have always used a frequentist approach due to our familiarity with it. We have contemplated the use of the Bayesian paradigm for the clinical trial but opted against it for a variety of reasons though we remain interested in this topic.

Reviewer: Dr Simon R White, University of Cambridge

Reviewer #3 (Remarks to the Author):

I found the study relevant and novel, and of interest to the academic diabetes community. The study achieved its primary outcome (HbA1c reduction), though I offer a few suggestions to

support the validity of the findings. My comments are below:

Results

1. HbA1c outcomes – suggest that for completeness within the text, that the authors also report the 95% confidence intervals or standard deviation of the within group differences (including in the abstract). I acknowledge that this is reported in the tables and figures.

Response: We thank the Reviewer for this comment. We have added the within group differences in the abstract.

2. Are there statistics available for the statements regarding the increased proportion of participants with improvements in HbA1c by 0.5%, 1.0% and 1.5%?

Response: We have added these statistics for the difference in proportions (p values and confidence intervals) to the revised manuscript (pages 5 and 9). We have also added these outcomes in Table 3.

3. Could the authors confirm that there was balance between groups for participants wearing their own Dexcom G6 (ie. concurrent CGM use, not just previous CGM use)? Participants had to manually enter their mealtime glucose level, and there was mention of a glucose reading discrepancy between the libre and Dexcom sensors (libre reading lower). Balance between groups becomes relevant where possibly, participants may have chosen to input their glucose level from the Dexcom rather than Libre, thereby affecting the insulin dose recommendation from the app.

Response: There was an exact balance (50%) in the number of participants who had concurrent Dexcom G6 CGM use in the experimental group and in the control groups. This has been added to the revised manuscript (page 5).

Note that we asked participants to input values from the study Freestyle libre sensors in the app. This has been added on page 14 for clarity.

Methods

4. There are 2 figures both labelled supplementary figure 5. In the methods, fig 5 is reference, though I assume is meant to refer to Fig 6.

Response: Thank you for pointing out this typo. It has been corrected with the updated Fig numbers.

5. Were the adaptive and non-adaptive groups given the same glucose target in the app? This detail may have been missed in my reading, or otherwise could be made more obvious

Response: We thank the Reviewer for this question. For both groups, the glucose target was set upon the app installation during the enrollment visit. All baseline insulin parameters, including the glucose target, were programmed based on their current treatment regimen per their health care provider's recommendations, and thus were different between participants. We have

clarified this detail in the text (page 14). Nevertheless, there was no difference in the mean baseline glucose target between control (6.7 mmol/L \pm 0.73) and experimental (6.5 mmol/L \pm 0.51) groups (page 4).

Endpoints

6. *Does 'insulin delivery' mean insulin dose?*

Response: Yes. This has been clarified in the text (pages 8 and 17).

Discussion

7. *Other RCTs assessing decision support systems for MDI users do not report improvements in glycemia. The discussion could be strengthened with some comment about possible reasons for the favourable result – differences in the McGill algorithm, or differences in the patient population compared to other studies?*

Response: We thank the Reviewer for this question. Although it is not possible to pinpoint the main reason for the favorable results in our study compared to other studies, the differences could be related to differences in the learning algorithms' design, the apps' design, the study designs, or the study populations, among others. A primary distinctive characteristic in our study is the inclusion of those with HbA1c \geq 7.5%. This is now stated in the revised manuscript (page 9).

Other minor comments:

8. *the supplementary tables are out of order, relative to where they are referenced in the text*

Response: We thank the Reviewer for raising this point. We have rearranged them per order mentioned in the text.

9. *Supp table 2 – could prandial insulin also be reported as Units/day, since basal is reported as units per day. It would be helpful to see this side by side.*

Response: We have added bolus insulin (U/day). Note that these insulin outcomes were added to Supplementary Table 2. Supplementary Table 4 continues to show the changes in insulin dose parameters from baseline to the last week.

10. *Could the statistics used to determine the p-value please be included as a footnote under each table.*

Response: We thank the Reviewer for this suggestion. The statistics have been included under each table.